# Physical Activity-Related Injuries and Risk Factors among Secondary School Students in Hong Kong

**DOI:** 10.3390/ijerph17030747

**Published:** 2020-01-24

**Authors:** Weicong Cai, Yan Sun, Ke Peng, Heather Kwok, Lin Lei, Shing Wu, Chi Kei Lam, Liping Li, Yang Gao

**Affiliations:** 1Department of Non-Communicable Disease Control and Prevention, Shenzhen Center for Chronic Disease Control, Shenzhen 518020, China; 16wccai@alumni.stu.edu.cn (W.C.); lin.leilana@gmail.com (L.L.); 2Injury Prevention Research Center, Shantou University Medical College, Shantou 515041, China; lpli@stu.edu.cn; 3Department of Sport and Physical Education, Hong Kong Baptist University, Hong Kong, China; sy7566048@163.com (Y.S.); heatherk@hkbu.edu.hk (H.K.); shingwu@hkbu.edu.hk (S.W.); cklam@hkbu.edu.hk (C.K.L.); 4The George Institute for Global Health, University of New South Wales, Sydney, NSW 2042, Australia; kpeng@georgeinstitute.org.au; 5School of Public Health, The University of Sydney, Sydney, NSW 2042, Australia

**Keywords:** physical activity, sports injury, recreational injury, risk factor, secondary school, adolescent

## Abstract

Increase in physical activity (PA) might elevate risks for physical activity-related injuries (PARIs). This study aimed to investigate the incidence rate and risk factors for PARIs among secondary school students in Hong Kong. All eligible students from three secondary schools were invited to participate in the study from November to December 2014. Information on PARI occurrences in the past 12 months, PA participation, and sociodemographics was collected. Multivariate and hierarchical logistic regression models were performed to estimate the risk of potential factors for PARIs. In total, 1916 students in forms 1–6 (aged 14.3 ± 1.7 years) completed valid questionnaires, with an overall yearly PARI incidence rate of 32.1% (boys: 34.3%; girls: 29.3%). There were sex differences in the risk factors identified, except that family size was associated with PARIs for all participants. Longer duration and higher intensity of PA increased the risk for PARI occurrences in both sexes. In conclusion, PARIs were prevalent among secondary school students in Hong Kong, and different sexes had different PARI incidence rates and were influenced by different risk factors. There is an urgent call for effective sex-specific interventions to prevent PARIs in this population.

## 1. Introduction

Physical activity (PA) is defined as any bodily movement produced by skeletal muscle that requires energy expenditure [1]. Physical inactivity has been identified as the fourth leading risk factor for global mortality, causing an estimated 3.2 million preventable deaths globally (6% of the total deaths) [1]. There is conclusive evidence that the physical fitness and health status of children and youth are substantially enhanced by frequent PA participation. Compared to inactive young people, physically active children and youth have higher levels of cardiorespiratory fitness, muscular endurance, and muscular strength, and the well-documented health benefits include reduced body fat, more favorable cardiovascular and metabolic disease risk profiles, enhanced bone health, and reduced symptoms of anxiety and depression [2,3,4].

Although the significant benefits of an active lifestyle are evident, it is also true that being physically active may elevate the risk for physical activity-related injuries (PARIs). The potential side effects of injury and fears of being injured have been barriers to adopting a more active lifestyle [5]. Internationally, PARIs account for 32.0%–55.0% of all injuries in boys and 19.0%–59.0% of all injuries in girls aged 11–15 years across eight western countries [6], indicating that PARI is a dominant category of child injury. In terms of the incidence of PARIs, previous studies in China revealed annual incidence rates of 29.9%–42.0% in boys in secondary school and 21.6%–25.0% in girls [7,8]. In addition to causing medical and financial burden, PARIs may lead to additional consequences for school children, such as class absence, barriers to PA participation, and even impacted physical growth [9,10,11]. It is estimated that up to 8% of children discontinue recreational sporting activities annually because of PARI [12]. Additionally, they might lose their enthusiasm for PA participation due to potentially negative perceptions and fear of injury [13].

In Hong Kong, the participation rate for PA is high (84.5%) among adolescents, while at the same time, only 8.4% of them meet the World Health Organization (WHO) recommendation [14]. Data from the Hong Kong Student Obesity Surveillance project among 32,005 adolescents aged 13–18 years revealed that more boys participated in structured and planned PA than girls, whilst the prevalence of non-exercise PA was similar between both sexes [15], suggesting that sex difference exists in PA patterns among Hong Kong adolescents. However, as far as we know, there is no study that has been focused on the PARI incidence rate among secondary school students in Hong Kong, and minimal research exists aiming to reveal its patterns and risk factors in other places. It is essential to understand the epidemic of PARIs and identify potential risk factors, which is also a prerequisite for protecting adolescents from PARIs. Therefore, we carried out a cross-sectional study to investigate the incidence rate and risk factors for PARIs among secondary school students in Hong Kong.

## 2. Materials and Methods

### 2.1. Study Participants

This study was conducted with secondary school students in Hong Kong from November to December 2014. There were 418,787 students studying in 519 secondary schools in that academic year. The majority of students were studying in 490 local schools, including government, aided, and direct subsidy scheme schools. Less than 6% were in 29 international schools [16]. Given significant differences existed between students in the two types of schools and the small proportion of international students, we recruited local school students only. A cluster sampling method was adopted, with study schools being recruited first, followed by all eligible students within those schools. Ethics approval was obtained from the Research Ethics Committee, Hong Kong Baptist University (reference number: FRG2/13-14/032). Students and their parents signed written informed consent forms in advance.

### 2.2. Data Collection

A self-administered questionnaire (Cronbach’s α = 0.818) was developed to collect data, including sociodemographic characteristics, PA habits, and PARI occurrences in the past 12 months. Students filled in the questionnaires in classrooms with assistance from our trained investigators after receiving a brief introduction to the study and questionnaire instructions.

Sociodemographic variables consisted of (1) student’s sex, age, place of birth, school, and grade; (2) parents’ education, occupation, employment status, and marital status; and (3) house type, house ownership, family size, family income, and car ownership.

PA, as an important factor related to PARI, was collected using the Children’s Leisure Activities Study Survey Chinese-version (CLASS-C), which had been proven to have good reliability in Hong Kong children aged 8–18 years old [17] and our previous study [7], and had been demonstrated to possess good internal consistency and one-week test-retest reliability in an earlier study [18]. The CLASS-C study collects information on 31 physical activities during leisure time, school physical education (PE) classes, and transportation in a typical week. For each activity, separate questions were asked regarding frequency and duration of participation on weekdays (Monday to Friday) and weekends (Saturday and Sunday). The intensity of each activity was classified according to a compendium of physical activities [19]. Daily time (min/day) spent in moderate- to vigorous-intensity PA (MVPA) and vigorous-intensity PA (VPA) was then calculated for each student. Participants were further divided into four groups according to their daily time of MVPA and VPA.

A PARI occurrence was defined as any injury suffered in the past 12 months that resulted from participating in PE classes, sports activities, transportation, or leisure time PA with one or more of the following consequences that meant that the student [20] (1) had to immediately stop the PA; (2) could not fully participate in the next planned PA; (3) could not go to school the next day; or (4) needed medical attention (e.g., from providers ranging from first aid personnel to general physicians or physiotherapists). Students were requested to report and count their PARI occurrences in the past 12 months according to the criteria. Respondents who reported PARI experiences were required to provide details of the latest PARI episode, including day and time, place, cause, mechanism, type, injured body part, PA involved, medical attention, days of class absence, time loss from PA, and other consequences of the injury. These detailed PARI characteristics helped to further validate the data of PARI incidence.

### 2.3. Data Analysis

Statistical Product and Service Solutions (SPSS) for Windows 23.0 (SPSS Inc., Chicago, IL, USA) was used to perform all data analyses. The person-based annual incidence rates of PARIs were estimated for boys, girls, and all students. Continuous and categorical variables were displayed as mean (standard deviation, SD) and number (percentage), and tested by independent sample *t* tests and chi-square tests, respectively, for between group differences. Both univariate and multivariate or hierarchical logistic regression models were performed to estimate the risks of sociodemographic and PA variables for PARIs, respectively, before and after controlling for each other. Unadjusted and adjusted Odds ratios (OR_unadj_ and OR_adj_) and 95% confidence intervals (95% CIs) were then derived. In the multivariate and hierarchical analyses, backward selection (Wald) was used to select variables using entry and removal criteria of *p* < 0.10 and *p* > 0.10. A significance level of 0.05 was adopted for all statistical tests.

## 3. Results

### 3.1. Participation

Of the 2645 eligible students in the three study schools, 1916 students participated in the study, with a participation rate of 72.4%. The participants aged 11 to 20 years, with a mean and SD of 14.3 and 1.7 years, respectively. The proportion of boys was slightly higher than that of girls (52.6% vs. 47.6%). There were few form 6 students (4.6%, the last year of secondary school) in the participants, as only one school granted us access to them. Over half of the students did not know their family monthly income (58.2%). Those students living in private housing and those living with 3 to 4 persons in a family represented a larger percentage compared to their counterparts.

### 3.2. PARI Occurrences

Six hundred and nine students suffered from PARIs in the past 12 months, with an annual person-based incidence rate of 32.1%. Of them, about two-fifths experienced PARI only once (19.0%) or twice (20.9%), while nearly one-quarter of the injured participants reported seven PARI occurrences or more. After being injured, most students stopped the PA immediately (67.1%) or could not participate into the next planned PA (48.3%), followed by visiting non-accident and emergency departments (28.6%), having school absence (23.3%), and receiving first aid on the spot (22.5%). Less than one-tenth of the injured stayed overnight in hospital (6.7%) or received hospitalization (7.7%).

### 3.3. Sociodemographic Characteristics and PARIs

Table 1 summarizes the PARI incidence rates by sociodemographic characteristics before and after stratification by sex. More than one-third of boys suffered from injuries, which was significantly higher than girls (34.3% vs. 29.3%, *p* = 0.020). Chi-square tests revealed significant differences in the injury distribution by mother’s education, occupation, employment status, and family size for boys, and almost all factors for girls except for the place of birth, father’s occupation, mother’s employment status, and marital status of parents. Considering there were significant differences in PARI incidence rate and its distribution by sociodemographic factors between boys and girls, further analyses were separately performed for each sex.

Table 2 presents the results of the univariate and multivariate logistic regression after stratification by sex. After controlling for other factors, boys whose mothers had high school education (OR_adj_ = 2.00, 95% CI: 1.35–2.97, *p* = 0.001), those whose mothers were housewives (OR_adj_ = 1.71, 95% CI: 1.17–2.50, *p* = 0.006), and those living in small-sized families (OR_adj_ = 1.59, 95% CI: 1.07–2.37, *p* = 0.021) were more likely to suffer from PARIs. In addition, increased risks approached, though did not reach, the significance level for maternal education level of tertiary and above (OR_adj_ = 1.59, 95% CI: 1.00–2.54, *p* = 0.052) and house ownership (OR_adj_ = 1.34, 95% CI: 0.98–1.84, *p* = 0.052). Among girls, school and grade were significantly related to PARIs. Compared to school A, girls in the other two schools were more likely to report injuries (OR_adj_ = 2.00, 95% CI: 1.35–2.96, *p* = 0.001 for School B and OR_adj_ = 1.87, 95% CI: 1.20–2.91, *p* = 0.006 for school C). Girls at higher grade levels were less likely to be injured, which reached significance for form 5 (OR_adj_ = 0.53, 95% CI: 0.32–0.86, *p* = 0.010) but not for form 6 (OR_adj_ = 0.49, 95% CI: 0.23–1.04, *p* = 0.065). Similarly to boys, small family size was also a risk factor for girls (OR_adj_ = 1.91, 95% CI: 1.23–2.96, *p* = 0.004). Moreover, girls from families owning private cars were 65% more likely to suffer from PARIs (OR_adj_ = 1.65, 95% CI: 1.19–2.28, *p* = 0.002).

### 3.4. PA and PARIs

More than half of the students (54.5%) failed to meet the WHO’s recommended PA level for children (i.e., MVPA < 60 min/day). About half (50.6%) performed VPA for less than 10 min/day. Girls were less active than boys, especially for VPA (% of MVPA < 60 min/day: 58.9% in girls vs. 40.7% in boys; % of VPA < 10 min/day: 61.6% in girls vs. 40.8% in boys). Distribution of the raw data for MVPA and VPA participation (min/week) is presented in Appendix A. As shown in Table 3, distinct increasing trends of PARI occurrences were observed along with the increases in the cumulative time of MVPA and VPA in both sexes, with the injury incidence rates in the highest categories (MVPA ≥ 90 min/day and VPA ≥ 50 min/day, respectively) being double those at the lowest levels (MVPA < 30 min/day and VPA < 10 min/day).

Table 4 displays the results of logistic regression before and after controlling for sociodemographic variables. Compared to the lowest PA levels, significantly elevated risks for PARIs were observed at all the other levels, except for the second quartile of MVPA for girls (MVPA of 30–59 min/day). Once again, a distinctly rising trend of the risk existed for each PA indicator and for both sexes, with the highest likelihood being found in the highest VPA category in boys (OR_adj_ = 3.41, 95% CI: 2.33–4.97, *p* < 0.001). Association between the involved sociodemographic factors and PARIs in the hierarchical models remained similar to those in the multivariate analyses in Table 2. The only exception was that of mother’s education in boys, where the risk for tertiary education level increased and reached significance (OR_adj_ = 1.76, 95% CI: 1.09–2.86, *p* = 0.022 for MVPA, OR_adj_ = 1.77, 95% CI: 1.09–2.87, *p* = 0.021 for VPA) compared to the middle school level and below.

In the hierarchical models, significant sociodemographic factors in the sex-specific multivariate analyses in Table 2 were adjusted for.

## 4. Discussion

In summary, about 32.1% of students who participated in this study experienced PARIs in the past 12 months. In terms of the frequency of PARIs, higher proportions were found at both ends of the spectrum (i.e., once or twice and more than seven times). The injuries resulted in a number of consequences, with the most frequent being stopping the PA immediately, followed by stopping the next planned PA, to staying overnight in a hospital. Boys had more injuries than girls. For boys, maternal education of high school, maternal occupation of housewives, and small family size were significantly related to PARIs, whereas school, grade, ownership of private cars, and small family size were significantly related to PARIs for girls. PARI risk increased with the cumulative time of MVPA and VPA in both boys and girls.

This study showed that nearly one-third (32.1%) of secondary school students in Hong Kong suffered from at least one PARI during the previous 12-month period. The incidence rate in this study is clearly higher than that (25.1%) reported by Cai et al. [7], but slightly lower than another report [8]. All these studies were cross-sectional designs, with the same working definition of PARIs. However, participants in these reports differed in age, with negative relationship between age and incidence rate being found in these studies. Thus, age may have contributed to the discrepancy in PARI incidence rate across the studies. In addition, different regions and time periods in data collection might also play a part in the discrepant findings. Nevertheless, we should put more emphasis on the problem of PARI and develop effective preventive programs to reduce the occurrence of PARI among secondary school students who are recommended to be physically active.

In line with earlier studies [7,8,9,21], the present study indicated that boys were more like to experience PARIs than girls. In comparison with girls, boys tend to be more physically active [7,22], which might increase their risk for PARIs. This is supported by our data that girls spent lower amounts of daily time performing both MVPA and VPA. In this study, PARIs of teenage girls were found to be more likely than boys to relate to school factors, such as schooling grade and the school itself. As shown in Appendix A, girls in schools B and C were more active than those in school A (median values MVPA: 300 min/day, 360 min/day, and 380 min/day in schools A, B, and C, respectively). Differences in the medians reached statistical significance in the independent samples median test (*p* = 0.04). Thus, the higher PA levels observed among girls in schools B and C may have at least partially contributed to the elevated risks for PARIs. Besides this, girls were more likely to be influenced by family factors, including parent’s education, occupation, and employment status. In contrast, of particular note was that the risk level for teenage boys was only related to maternal factors. Paternal indicators were found to have no association with the PARIs for boys. As indicated, school and family factors were related to the PARI occurrences differently in the different sexes. This study provides evidence of the importance of taking school and family into account in developing future injury prevention measures.

Small family size appears to be a risk factor of PARIs for teenagers. Teenage girls and boys in a family size of one to two were 91% and 59% more likely to experience PARI than in a family size of three to four. A family size of one to two essentially translates into living alone or in a single-parent family. Therefore, this study indicates that adolescents raised in a single-parent family are more likely to suffer from PARIs. Earlier studies found that children living with single parents might have an elevated risk of developing physical health, behavioral, and sociopsychological problems and injury compared with those in two-parent families [23,24,25]. Furthermore, the absence of parental injury concerns might also be related to a higher risk for PARIs [5]. Thus, more attention should be paid to children from single-parent families in PARI prevention and related programs.

The socioeconomic status of the children’s families may be a risk factor for PARIs. In this study, affluent families were indicated by several family factors (e.g., parental education, parental occupation, car ownership, house ownership, and family income). This finding is consistent with other studies [23] and could possibly be explained by two factors. Firstly, parents with higher education level are aware of the benefits of PA, and therefore encourage their children to exercise more. Secondly, performing PA is more affordable for rich families. In the present study, housing ownership and car ownership showed the highest correlation with risk due to family affluence in boys and girls, respectively. It is reported that boys and girls with house or car ownership have higher PARI rates, with the difference being insignificant in boys and significant in girls. Future studies could focus on these contributors to identify the mechanism of injury, especially for girls.

Another risk factor for PARIs is the MVPA and VPA levels. The higher the two levels, the higher the PARI incidence rate. This finding is consistent with previous studies showing that longer duration and higher intensity of PA participation would increase the risk for PARIs [7,8,26]. Several PA recommendations exist for frequency, duration, and intensity, but few have been developed from a safety perspective. It is obvious that there is a knowledge gap in PA guidelines that necessitates further study. These highly active adolescents are the priority in future intervention actions to reduce the PARI occurrences when promoting a physically active lifestyle.

Some limitations should be noted. Firstly, the use of a self-reported questionnaire to measure the variables is subject to recall bias, particularly in the questions capturing the number of injuries in the past 12 months and the subsequent actions taken. Students might not have reported accurate information, for example underreporting the number of injuries (particularly minor injuries). As a result, the incidence rate might be underestimated. Despite this limitation, having the students self-report was the most feasible, practical, and cost-effective means to obtain the information, especially as this study has a large sample size. The second limitation of this study is the study design. The cross-sectional study design cannot provide a cause-and-effect relationship amongst the variables. The analyses of the variables, such as maternal education and PARI rate, merely implied an association. Nevertheless, the association is not proven until a further confirmatory study is completed. Another drawback of this study is generalizability and representativeness. The study samples only include secondary school students. Hence, the results could not be extended to children of other ages [27]. In addition, the use of the cluster sampling method and the fact that only three study schools were recruited led to poor representativeness.

## 5. Conclusions

PARIs are prevalent in secondary school students in Hong Kong. The PARI incidence rate is positively associated with the time spent performing MVPA and VPA, and a stronger association could be found with VPA than MVPA. Most risk factors of PARIs differ between boys and girls, with more school and family factors being identified in girls. Regardless of sex, those students living alone or with a single parent are more likely to experience PARIs. The results in this study require further studies for confirmation. Nevertheless, these findings give insights into the importance of PARI prevention when advocating a physically active lifestyle, particularly for adolescents who are performing high-intensity PA for longer durations, growing up in single-parent families, and living in affluent families.

## Figures and Tables

**Table 1 ijerph-17-00747-t001:** Physical activity-related injury (PARI) incidence rate (%) by sociodemographic characteristics of the participants.

Characteristics	All		Boys		Girls	
	Student No.	PARI Rate (%)	Student No.	PARI Rate (%)	Student No.	PARI Rate (%)
School		***p*** **< 0.001**		*p* = 0.205		***p*** **< 0.001**
A	675	26.2	360	31.5	315	20.3
B	694	36.5	347	37.9	342	34.5
C	547	33.5	297	33.3	248	33.5
Grade		*p* = 0.056		*p* = 0.389		***p*** = **0.039**
F1	410	33.3	209	33.7	196	31.8
F2	347	30.9	163	28.8	182	32.6
F3	341	38.1	197	39.6	144	36.1
F4	366	31.7	216	35.3	150	26.8
F5	364	27.4	181	34.1	183	20.9
F6	88	27.3	38	28.9	50	26.0
Place of birth		*p* = 0.050		*p* = 0.175		*p* = 0.219
Hong Kong	1680	32.7	899	34.7	775	30.1
Other	163	25.2	71	26.8	92	23.9
Paternal education		*p* = 0.099		*p* = 0.658		***p*** = **0.029**
≤Middle school	464	28.3	247	33.6	215	22.1
High school	602	34.3	314	36.7	286	31.3
≥Tertiary	763	32.9	399	33.7	361	31.9
Paternal occupation		***p*** = **0.025**		*p* = 0.534		*p* = 0.055
Administers + Professionals	1026	34.5	545	36.0	475	33.1
Clerks + Sales	251	26.8	126	29.4	125	24.2
Workers	424	28.7	215	33.2	209	24.2
Others	215	31.0	118	33.6	96	28.1
Paternal employment status		*p* = 0.112		*p* = 0.963		***p*** = **0.024**
Full-time	1618	32.6	847	34.2	765	30.4
Others	252	27.5	132	34.4	119	20.3
Maternal education		***p*** **< 0.001**		***p*** = **0.007**		***p*** < **0.001**
≤Middle school	453	23.2	235	27.5	217	18.7
High school	758	36.0	407	39.4	348	31.6
≥Tertiary	622	33.4	316	32.7	303	33.8
Maternal occupation		*p* = 0.139		***p*** = **0.043**		***p*** = **0.034**
Administers + Professionals	600	33.9	316	31.5	279	36.2
Clerks + Sales	455	29.0	227	30.4	227	27.3
Workers	80	23.1	42	26.8	38	18.9
Housewives	705	33.7	375	40.3	329	26.1
Others	76	29.3	44	30.2	32	28.1
Maternal employment status		*p* = 0.458		***p*** = **0.042**		*p* = 0.267
Full-time	996	31.4	505	31.4	485	31.0
Others	888	33.0	478	37.6	409	27.6
Marital status		*p* = 0.406		*p* = 0.468		*p* = 0.759
Married	1724	32.1	908	34.4	809	29.3
Others	159	28.9	76	30.3	83	27.7
Housing type		***p*** = **0.006**		*p* = 0.111		***p*** = **0.001**
Public	662	27.2	357	32.7	305	20.9
Government-aided	170	34.9	88	40.2	81	28.4
Private	905	33.5	449	32.3	450	34.2
Others	151	39.7	92	43.5	59	33.9
House ownership		***p*** = **0.004**		*p* = 0.055		***p*** = **0.037**
No	875	28.4	451	30.5	421	25.9
Yes	957	34.7	505	36.5	448	32.4
Family size		***p*** **< 0.001**		***p*** = **0.008**		***p*** = **0.013**
1–2 persons	293	41.3	164	42.5	129	39.8
3–4 persons	896	29.7	454	31.1	440	27.9
>4 persons	553	28.4	283	31.3	268	25.1
Missing	174	40.7	103	43.6	68	35.3
Car ownership		***p*** **< 0.001**		*p* = 0.219		***p*** **< 0.001**
No	1071	27.9	569	32.1	500	22.9
Yes	778	36.3	392	36.0	382	36.7
Family monthly income (HK$)		***p*** = **0.047**		*p* = 0.346		***p*** = **0.019**
≤20,000	300	27.0	166	30.9	132	21.2
20,001–40,000	194	37.5	110	41.3	84	32.5
>40,000	293	35.6	172	33.3	121	38.8
Do not know	1097	31.4	534	33.9	558	28.8
CSSA recipient		*p* = 0.060		*p* = 0.510		***p*** = **0.002**
No	1719	32.4	930	33.9	809	30.5
Yes	147	24.8	50	37.9	80	13.9

Missing <5% was ignored and not presented here. CSSA: Comprehensive Social Security Assistance (CSSA) scheme. The CSSA Scheme provides a safety net for those who cannot support themselves financially. It is designed to bring their income up to a prescribed level to meet their basic needs. Supplements consist of long-term supplement, single-parent supplement, community living supplement, transport supplement, and residential care supplement. The bold indicates there are statistically significant differences.

**Table 2 ijerph-17-00747-t002:** Risk estimates of sociodemographic characteristics for PARIs by sex.

Variables	Univariate Model		Multivariate Model	
	OR_unadj_	(95% CI)	OR_adj_	(95% CI)
**BOYS**				
Maternal education				
≤Middle school	1.00		1.00	
High school	1.72 **	(1.21, 2.44)	2.00 **	(1.35, 2.97)
≥Tertiary	1.28	(0.89, 1.86)	1.59 ^#^	(1.00, 2.54)
Maternal occupation				
Administers + Professionals	1.00		1.00	
Clerks + Sales	0.95	(0.66, 1.37)	0.91	(0.60, 1.38)
Workers	0.80	(0.38, 1.65)	1.09	(0.48, 2.47)
Housewives	1.47 *	(1.07, 2.01)	1.71 **	(1.17, 2.50)
Others	0.94	(0.47, 1.88)	0.83	(0.28, 2.46)
House ownership				
No	1.00		1.00	
Yes	1.31 ^#^	(0.99, 1.72)	1.34 ^#^	(0.98, 1.84)
Family size				
3–4 persons	1.00		1.00	
1–2 persons	1.64 ***	(1.13, 2.37)	1.59 *	(1.07, 2.37)
>4 persons	0.62	(0.41, 0.92)	0.89	(0.63, 1.25)
Missing	1.71 *	(1.10, 2.66)	1.64 ^#^	(0.98, 1.84)
**GIRLS**				
School				
A	1.00		1.00	
B	2.07 ***	(1.45, 2.96)	2.00 **	(1.35, 2.96)
C	1.98 ***	(1.35, 2.90)	1.87 **	(1.20, 2.91)
Grade				
F1	1.00		1.00	
F2	1.04	(0.67, 1.60)	0.87	(0.55, 1.39)
F3	1.21	(0.77, 1.91)	1.22	(0.76, 1.95)
F4	0.79	(0.49, 1.26)	0.68	(0.42, 1.12)
F5	0.57 *	(0.35, 0.90)	0.53 *	(0.32, 0.86)
F6	0.75	(0.37, 1.52)	0.49 ^#^	(0.23, 1.04)
Family size				
3–4 persons	1.00		1.00	
1–2 persons	1.72 *	(1.14, 2.59)	1.91 **	(1.23, 2.96)
>4 persons	0.87	(0.61, 1.23)	0.75	(0.52, 1.08)
Missing	1.41	(0.82, 2.42)	1.56	(0.87, 2.82)
Car ownership				
No	1.00		1.00	
Yes	1.95 ***	(1.45, 2.62)	1.65 **	(1.19, 2.28)

Note: ^#^
*p* < 0.10; * *p* < 0.05; ** *p* < 0.01; *** *p* < 0.001.

**Table 3 ijerph-17-00747-t003:** Distribution of physical activity (PA) level and PARI occurrences among participants by sex.

Variables	All		Boys		Girls	
	Student No.	PARI Rate (%)	Student No.	PARI Rate (%)	Student No.	PARI Rate (%)
MVPA level		*p* < 0.001		*p* < 0.001		*p* < 0.001
<30 min/day	477	21.1	227	21.6	250	20.6
30–59 min/day	530	26.9	261	30.7	267	22.6
60–89 min/day	360	36.9	203	40.1	156	32.9
≥90 min/day	480	46.0	271	45.0	205	46.8
VPA level		*p* < 0.001		*p* < 0.001		*p* < 0.001
<10 min/day	947	23.1	401	23.9	545	22.5
10–29 min/day	377	35.3	220	36.4	157	33.8
30–49 min/day	219	40.0	134	38.2	82	40.7
≥50 min/day	329	50.3	225	49.5	101	51.5

Missing data <5% was ignored and not presented here.

**Table 4 ijerph-17-00747-t004:** Risk estimates of PA levels for PARIs by sex.

Variables	Univariate Model		Hierarchical Model	
	OR_unadj_	(95% CI)	OR_adj_	(95% CI)
**BOYS**				
MVPA level				
<30 min/day	1.00		1.00	
30–59 min/day	1.55 *	(1.03, 2.32)	1.81 **	(1.17, 2.81)
60–89 min/day	2.34 ***	(1.54, 3.54)	2.43 ***	(1.54, 3.82)
≥90 min/day	2.85 ***	(1.94, 4.21)	3.25 ***	(2.12, 4.97)
VPA level				
<10 min/day	1.00		1.00	
10–29 min/day	1.83 **	(1.27, 2.62)	1.88 **	(1.28, 2.77)
30–49 min/day	1.97 **	(1.29, 3.00)	2.03 **	(1.29, 3.19)
≥50 min/day	3.13 ***	(2.21, 4.44)	3.41 ***	(2.33, 4.97)
**GIRLS**				
MVPA level				
<30 min/day	1.00		1.00	
30–59 min/day	1.11	(0.73, 1.69)	1.02	(0.65, 1.58)
60–89 min/day	1.87 **	(1.19, 2.94)	1.62 *	(1.01, 2.60)
≥90 min/day	3.36 ***	(2.23, 5.05)	2.86 ***	(1.84, 4.42)
VPA level				
<10 min/day	1.00		1.00	
10–29 min/day	1.75 **	(1.19, 2.58)	1.67 *	(1.10, 2.51)
30–49 min/day	2.37 **	(1.45, 3.85)	2.25 **	(1.34, 3.78)
≥50 min/day	3.65 ***	(2.36, 5.67)	3.09 ***	(1.93, 4.94)

Note: * *p* < 0.05; ** *p* < 0.01; *** *p* < 0.001.

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
