# Peer review of "Physical Activity-Related Injuries and Risk Factors among Secondary School Students in Hong Kong"

_ijerph, 2020, doi:10.3390/ijerph17030747_

Round 1

Reviewer 1 Report

Introduction

1 It is unnecessary to refer to the recommendations WHO in the introduction, because it adds nothing to the topic about the risk factors of injury during activity ??
2. There is lack of a defined objective or objectives, what's does not allow understanding of the authors' intentions
a) do they want to identify physical activity related injuries among pupils?
b) do they want to assess influence of incidence injuries on frequency and intensify for physical activity among pupils ??
c) and or both - but is lack this information in the introduction there

Methodology
There is not possible to identity  whether the survey questionnaire was validated and how consistently it was applied. By whom and when the data was collected are factors that may influence the quality of the data. How the questions of the protocol were translated to language other than English. I think that it is necessary to include questionnaire as Appendix.

Results
line 47: what is the difference between injury and incidence, as the authors define  injury and incidence. This is particularly true - line 130: Less than one tenth of the injured stayed overnight in hospital (6.7%) or received hospitalization (7.7%) - should they be included to statistic? line 157: what does it means : nearly reached significance . statistical significance or not significance. Authors should be precise.

Discussion
1) This section lacks an attempt to interpret why the relationships occurred and what can be done, in practical terms, to minimize the risk of those injuries.
2) some threads should be improved due to remarks mentioned above. Cited literature is up to date

Reviewer 2 Report

This paper by Cai and colleagues is an analysis of retrospective physical activity and physical activity-related injuries, as well as demographic risk factors. I have some specific comments, which follow.

Gender vs. sex

As there is no questionnaire attached, I am assuming because it is not in English, I do not know whether you inquired about gender or sex. However, since you only report boys and girls, I assume it was sex. Yet you talk about gender in the abstract and throughout the paper. If this should be sex, please correct. If you in fact only asked about gender, I would suggest including the question in your methods so this is clear to the reader.

INTRODUCTION

Line 42

The sentence starting “Knowledge of these benefits of PA…” seems to be missing words.

Line 48

“concerns of resulting injury” does not fit in to this sentence. I understand what you are trying to say but please revise the grammar.

Lines 55-56

You are referencing a paper by Emery et al. on this when in fact this is from the Introduction on that paper, making it a secondary preference. Please retrieve the original reference and reference that.

Lines 63-64

What does “no available study” mean? There is research but it is not available to you? Or there is no research? And if there is no research on this topic, how can there be less research on patterns and risk factors?

MATERIALS AND METHODS

It is unclear to me why the number of all students in secondary students is mentioned here as it does not seem to have anything to do with your study size or sampling.

How was the sample size arrived at? Were there only three schools in the area and they all consented? Or were the schools that declined to participate?

Were the PARI questions validated?

Line 98-103

It is unclear to me what you are referencing here with [20-22]. If you derived the criteria from different studies, please add the reference immediately after the criteria.

How did you decide to use P<0.10 and P>0.10 as entry and removal criteria?

RESULTS

Line 118

Were participation rates same in all three schools?

Line 119

“Boys were slightly more than girls…” Please revise this sentence.

Line 121

“… granted us to access to them” Please revise the grammar.

Line 122

“More students were..” Compared to what?

DISCUSSION

Line 186

Instead of “studied” I would say “students who participated” or something alone those lines.

Line 190

Why is there a sudden switch from “boys” and “girls” to “males” and “females”?

Line 197

You state “other reports” but reference only one.

Lines 233-238

I think you need to dig a bit deeper here to the association between PA participation and risk of injuries. I don’t think you can conclude from your results that those who participate more are “not attentive to the PARIs or do not know how to protect themselves from injury.” Also, I would recommend reading Emery et al. 2006. Injury prevention in child and adolescent sport: whose responsibility is it?

There were statistically significant differences in PARI rate between the schools. Why is that? What was different about the one school? This is not discussed.

Overall, the limitations of this study are brought to light and conclusions are supported by the findings. However, more clarity regarding the writing is needed as well as digging a bit deeper in the literature in the Discussion.  
